# Analyzing Olfactory Neuron Precursors Non-Invasively Isolated through NADH FLIM as a Potential Tool to Study Oxidative Stress in Alzheimer’s Disease

**DOI:** 10.3390/ijms22126311

**Published:** 2021-06-12

**Authors:** Laura Gómez-Virgilio, Alejandro Luarte, Daniela P. Ponce, Bárbara A. Bruna, María I. Behrens

**Affiliations:** 1Centro de Investigación Clínica Avanzada, Facultad de Medicina and Hospital Clínico Universidad de Chile, Santiago 8380453, CP, Chile; jalim166@gmail.com (L.G.-V.); aluarte@bni.cl (A.L.); dponcedelavega@gmail.com (D.P.P.); bbruna@ug.uchile.cl (B.A.B.); 2Departamento de Neurociencia, Facultad de Medicina, Universidad de Chile, Santiago 8380453, CP, Chile; 3Faculty of Medicine, Biomedical Neuroscience Institute, Universidad de Chile, Santiago 8380453, CP, Chile; 4Departamento de Neurología y Psiquiatría, Clínica Alemana de Santiago y Universidad del Desarrollo, Santiago 7650729, CP, Chile; 5Departamento de Neurología y Neurocirugía, Hospital Clínico Universidad de Chile, Santiago 8380453, CP, Chile

**Keywords:** oxidative stress, FLIM, Alzheimer’s disease

## Abstract

Among all the proposed pathogenic mechanisms to understand the etiology of Alzheimer’s disease (AD), increased oxidative stress seems to be a robust and early disease feature where many of those hypotheses converge. However, despite the significant lines of evidence accumulated, an effective diagnosis and treatment of AD are not yet available. This limitation might be partially explained by the use of cellular and animal models that recapitulate partial aspects of the disease and do not account for the particular biology of patients. As such, cultures of patient-derived cells of peripheral origin may provide a convenient solution for this problem. Peripheral cells of neuronal lineage such as olfactory neuronal precursors (ONPs) can be easily cultured through non-invasive isolation, reproducing AD-related oxidative stress. Interestingly, the autofluorescence of key metabolic cofactors such as reduced nicotinamide adenine dinucleotide (NADH) can be highly correlated with the oxidative state and antioxidant capacity of cells in a non-destructive and label-free manner. In particular, imaging NADH through fluorescence lifetime imaging microscopy (FLIM) has greatly improved the sensitivity in detecting oxidative shifts with minimal intervention to cell physiology. Here, we discuss the translational potential of analyzing patient-derived ONPs non-invasively isolated through NADH FLIM to reveal AD-related oxidative stress. We believe this approach may potentially accelerate the discovery of effective antioxidant therapies and contribute to early diagnosis and personalized monitoring of this devastating disease.

## 1. Introduction

Alzheimer’s disease (AD) is the most common cause of dementia and the sixth cause of death in the world, constituting a major health problem for aging societies [1]. This disease is a neurodegenerative continuum with well-established pathology hallmarks, namely the deposition of amyloid-β (Aβ) peptides in extracellular plaques and intracellular hyperphosphorylated forms of the microtubule associated protein tau forming neurofibrillary tangles (NFTs), accompanied by neuronal and synaptic loss [2]. Interestingly, patients who will eventually develop AD manifest brain pathology decades before clinical symptoms appear [3,4]. Nevertheless, AD is still frequently diagnosed when symptoms are highly disabling and yet there is no satisfactory treatment.

Although the manifestations of AD are preponderantly cerebral, cumulative evidence shows that AD is a systemic disorder [5]. Accordingly, molecular changes associated with AD are not exclusively manifested in the brain but include cells from different parts of the body, ranging from the blood and skin to peripheral olfactory cells. More recently, neurons derived from induced pluripotent stem cells (iPSCs) from AD patients have contributed to glean a more realistic insight of brain pathogenic mechanisms [6]. Alternatively, the culture of olfactory neuronal precursors (ONPs) has emerged as a relatively simpler tool to study different brain disorders, taking advantage of their neuronal lineage and their readily non-invasive isolation [7,8]. For instance, patient-derived ONPs manifest abnormal amyloid components together with tau hyperphosphorylation, which have recently led to the proposal of these cells as a novel diagnostic tool for AD [9,10,11].

Different hypotheses have attempted to explain AD pathogenesis. Some of them include Aβ cascade, tau hyperphosphorylation, mitochondrial damage, endoplasmic reticulum (ER) stress, and oxidative stress. Interestingly, although it has been difficult to establish a prevailing causative mechanism, increased levels of oxidative stress seem to be a common feature for many of these models. Furthermore, oxidative stress due to increased levels of reactive oxygen species (ROS) has been broadly recognized as a very early signature during the course of AD [12,13,14]. Interestingly, AD-related oxidative stress is by no means restricted to neuronal cells but is also related to astrocytes’ oxidative damage and antioxidant capacity [15]. Indeed, since the acknowledgment of the tripartite synapse, it has become increasingly clear that different antioxidant mechanisms of astrocytes can be harnessed by synaptically active neurons and surrounding cells [16,17,18]. In the tripartite synapse, the astrocyte’s endfeet are close to synapses and can be activated by the spillover of synaptic glutamate to provide a timely antioxidant response [19,20]. Moreover, it is not entirely understood how other glial cells such as pericytes may contribute to the damage induced by AD-related oxidative stress. For instance, oxidative damage may compromise the integrity of pericytes, which in turn could alter the blood-brain barrier’s integrity, favoring the infiltration of cytotoxic cells and the emergence of brain edema [21,22]. In coherence with a broader systemic manifestation of this disease, the peripheral olfactory system shows AD-associated oxidative stress, which has been measured both in the olfactory neuroepithelium and in cultured ONPs [23,24,25]. However, while the intriguing relationship between oxidative stress and AD has been long known, their translational impact has remained limited.

Interestingly, the oxidative status of cells is highly correlated with the content of autofluorescent metabolic co-factors such as NADH and its phosphorylated version NADPH [26,27,28,29]. In addition, NADH is required to synthesize NADPH, which is at the core of the antioxidant response of different cells by sustaining the synthesis of antioxidants such as glutathione (GSH) and thioredoxin [30]. Furthermore, it has been shown in AD animal models that the provision of NADH is upstream the levels of GSH in order to counterbalance increased ROS levels and neuronal death [27]. Interestingly, external manipulation of oxidative or reducing conditions of cultured neurons are directly manifested as changes in mitochondrial and cytosolic NADH content [28]. As such, by imaging NADH autofluorescence, it might be possible to obtain a real-time monitoring of redox imbalance without the need to use exogenous staining or recombinant sensors. Complementary to methodologies purely based on fluorescence intensity, Fluorescence Lifetime Imaging Microscopy (FLIM) has received increasing attention [31,32]. Fluorescence lifetime is the average time in which a fluorophore remains excited to emit photons before descending to the ground state, providing unique information about its biochemical environment. Importantly, NADH FLIM can be harnessed to increase the sensitivity to its autofluorescence and to discriminate its binding to enzymes from different signaling pathways. In this review, we explore the idea of using ONPs non-invasively isolated coupled to NADH FLIM to reveal AD-associated oxidative stress. This approach may have a broad impact for early AD diagnosis and treatment.

## 2. Olfactory Neuroepithelium and the Non-Invasive Isolation of ONPs

The olfactory neuroepithelium is a key structure for odor sensing. It consists of a pseudostratified columnar epithelium located on the outer domain of the olfactory mucosa settled on the basement membrane (BM) and the lamina propria (LP) [33]. The cellular composition of these layers has been widely documented based on morphological analysis and the use of characteristic markers for each cell type [34,35,36,37]. Figure 1 schematizes the location, cellular components, and molecular markers of the human olfactory mucosa.

The olfactory neuroepithelium is also a source of stem cells, which are capable of self-renewal and can generate neuronal precursors throughout the entire human lifetime. These precursors include neural stem cells known as basal cells. As expected for neural stem cells, basal cells are multipotent and allow the continuous replacement of neuronal and non-neuronal cells such as olfactory receptor neurons (ORNs) and sustentacular cells (of astrocytic lineage), respectively [38,39,40]. In addition, the LP contains another less accessible population of stem cells, whose features meet most of the minimum criteria of the mesenchymal and Tissue Stem Cell Committee of the International Society for Cellular Therapy [41]. As such, they are named as olfactory ectomesenchymal stem cells (OE-MSCs) [42,43,44].

Isolation of cells of the olfactory neuroepithelium from patients provides a source of cultured neural stem cells, which has been used to model different brain disorders such as schizophrenia, Parkinson’s disease, autism, ataxia-telangiectasia, hereditary spastic paraplegia (HSP), and AD [7,45,46,47,48,49]. These neural stem cells can be frozen and stored for subsequent use and tolerate several passages without significantly losing their main properties. Furthermore, purified cultures obtained by cloning selection through limiting dilution significantly increases cell viability at least until passage 60 [50]. In this work, we will refer to neural stem cells isolated from the olfactory neuroepithelium as olfactory neuronal precursors (ONPs), similar to [8,9,50,51].

Different strategies have been used to isolate and culture patient-derived ONPs, ranging from biopsies to non-invasive exfoliation of the nasal turbinate. Human ONPs were first isolated by Wolozin et al. from the olfactory neuroepithelium of cadavers or from adult biopsied samples [10,52]. Another similar isolation approach demonstrated that a significant subpopulation of these cells express markers of mature olfactory neurons such as OMP, Golf, NCAM, and NST and look small and bright to the microscope, in contrast to the remaining “dark phase” cells that do not express OMP, but glial markers [53]. However, a systematic characterization of these cultures has shown that after a few days in vitro, both dark and bright phase cells show an intracellular calcium increase in response to odorants, highlighting the neuronal features of these cells [54]. In addition, cells with features of ONPs have also been obtained from dissociated neurospheres, which have been denominated “olfactory neurosphere-derived” (ONS) cells [43]. Alternatively, ONPs can be non-invasively isolated by an exfoliation of the nasal cavity [51]. These exfoliated cells can be cultured in a modified media to propitiate neural lineage maintenance and proliferation. Notably, these neuronal precursors conserve their capability to differentiate into ORNs in the presence of dibutyryl adenosine 3’,5’-cyclic monophosphate (Db-cAMP) and, strikingly, maintain their electrical response to odorants [51]. Thus, non-invasively isolated ONPs retain neuronal features similar to those obtained by biopsy. A simplified extraction protocol and the molecular characterization of non-invasively isolated ONPs is shown in Figure 2.

## 3. Alzheimer’s Disease-Related Oxidative Stress in the Olfactory Epithelium and ONPs

Oxidative stress is the result of an imbalance between oxidant and antioxidant cellular pathways. One of the most studied oxidant compounds are ROS, which are highly reactive molecules, including peroxide (H_2_O_2_), superoxide anion radical (O_2_ • −), and hydroxyl radical (• OH), among others. These molecules may covalently interact with lipids, proteins, and carbohydrates, generating molecular adducts and cumulative damage that, when sensed by cells, may actively trigger different death programs [55].

It was well established almost three decades ago that oxidative stress damage is linked to AD [14]. Furthermore, it has been proposed that oxidative stress at different brain neuronal and non-neuronal cells might be the earliest event of a pathogenic cascade [13]. Whether oxidative stress is a causative agent or just a consequence in neurodegenerative disorders has been thoroughly debated for several years, but still remains an open question [56,57,58]. The most parsimonious interpretation of this evidence is that oxidative stress as well as other potential AD causative agents (such as Aβ accumulation) are part of a highly interconnected vicious cycle rather than a linear chain of events with a unique origin. The molecular mechanisms and implications of oxidative stress on the nervous system and, potentially, during AD pathogenesis have been thoroughly reviewed elsewhere [12,59]. Here, we focus on evidence showing AD-associated oxidative stress in the peripheral olfactory system rather than reviewing mechanistic explanations.

Oxidative stress associated with AD is manifested in the olfactory neuroepithelium. Accordingly, increased immunoreactivity of the antioxidant enzyme manganese and Copper-Zinc superoxide dismutases have been detected in ORNs and basal and sustentacular cells of the olfactory neuroepithelium of AD patients compared with age-matched controls [60]. Analogously, AD patients harbor a higher immunoreactivity against the antioxidant protein Metallothionein both in the olfactory neuroepithelium and the Bowman’s Glands and the LP [61]. Both results suggest that cells from olfactory neuroepithelium elicit an increased antioxidant defense, due to increased oxidative stress during AD. With respect to the direct measurement of oxidation products, post-mortem staining showed an increase in 3-nitrotyrosine (3-NT) in the brain and olfactory neuroepithelium of AD patients [23]. Figure 3 schematizes the antioxidant response and oxidative damage reported in ONPs and OE from AD patients. It would be of interest to uncover whether some AD genetic factors such as the *ApoE* ε4 allele (*ApoE4*) (the single most important genetic risk factor for AD) also manifests oxidative stress signatures in the olfactory epithelium. It is plausible that this is the case because deficits in odor fluency, identification, recognition memory, and odor threshold sensitivity have been associated with the inheritance of the *ApoE4* genotype in several studies [62,63,64]. For a more thorough compiling of evidence showing AD-associated oxidative damage across other domains of the nervous system, readers may refer to the following excellent articles [12,59,65].

The relationship between oxidative stress and AD has been extensively studied mainly through cellular and animal models [47,54]. However, these models may not fully capture key features of the disease. This limitation potentially leads to wrong conclusions about the pathogenic mechanisms and ultimately may dampen the development of effective therapies. Alternatively, patient-derived cells of neuronal lineage such as those from the olfactory epithelium may provide a convenient solution to this problem [5,9,42].

Interestingly, cultured patient-derived ONPs and other peripheral cells also manifest AD-associated oxidative stress. For example, an increase in the level of hydroxynonenal and Nɛ-(carboxymethyl)lysine) (indicating lipid peroxidation), as well as a higher content of heme oxygenase-1, has been found in ONPs isolated from AD patients compared with age-matched controls (Figure 3) [24]. Furthermore, ONPs from AD patients are also more susceptible to oxidative stress-induced cell death [25]. This is strikingly similar to what has been found by our group in blood-derived lymphocytes from AD patients [66,67]. Indeed, manifestations of oxidative stress associated with AD have been reported in different patient-derived peripheral cells ranging from blood cells to fibroblasts and iPSCs-derived neurons. These changes may include compensatory antioxidant responses and a rise in the concentration of oxidation by-products, as well as increased susceptibility to ROS-induced cell death, which has been demonstrated in different cellular types from AD patients. Many of those findings are summarized in the Table 1. In addition, Table 1 also summarizes similar evidence of other relevant pathogenic mechanisms proposed for AD pathogenesis, including Amyloid/Tau, mitochondria, and ER-stress. Thus, different cells throughout the body show signs of different proposed AD pathogenic mechanisms, including oxidative stress at early stages of the disease continuum. The robustness of this tendency highlights the potential of patient-derived cells, and in particular ONPs, for monitoring oxidative stress associated with AD.

## 4. The Role of NADH in Cell Metabolism and Antioxidant Defense

Metabolism is intimately associated with oxidative stress, since ATP production by mitochondria requires the reduction of oxygen to water, which is a major source of ROS. Enzymatic cofactors of energetic metabolism such as oxidized and reduced NAD (NAD+ and NADH, respectively), as well as their phosphorylated versions (NADP+ and NADPH), constitute key bridges between energy supply and the antioxidant defense of cells [30]. The availability of these cofactors is highly inter-related, and depending on the cellular context, their separate or combined measurement can be used to reveal redox homeostasis both in the cytosol and mitochondria [99]. We provide a brief overview of the main cellular sources and consumers of NAD+/NADH and their interplay with NADP+/NADPH levels with a special focus on neuronal cells.

The provision of NAD+ molecules in the body comes from de novo synthesis from tryptophan or via salvage pathways using nicotinamide (NAM) and nicotinamide riboside (NR) as precursors. The detailed pathways of NAD+ direct synthesis have been reviewed elsewhere [100]. In addition, the direct consumption of NAD+ is achieved mainly by the enzymatic activity of silent information regulator proteins or sirtuins (SIRTs) and poly (adenosine diphosphate-ribose) polymerases (PARPs). Sirtuins catalyze the deacetylation of target proteins by converting NAD+ into NAM and a *O*-Acyl ATP ribose. The activity of SIRTs has been profusely studied in the nucleus, where they control the function of different transcription factors and histone proteins to regulate cell senescence and neurodegeneration [101,102]. In addition, PARPs are enzymes that normally control DNA repair, whose overactivation under intense DNA oxidative damage may lead to cellular depletion of NAD+ and ATP. Both processes may promote cell death, potentially contributing to the pathogenesis of neurodegenerative disorders such as AD [103].

Different metabolic reactions determine the level and subcellular distribution of NADH. Accordingly, the synthesis of NADH from NAD+ in the cytosol is achieved by the glycolytic pathway, which generates two ATPs, two NADH, and two pyruvates as net yield per glucose. In addition, NADH is synthesized by two mitochondrial enzymes: pyruvate dehydrogenase (PDH), which produces acetyl-CoA entering to the tricarboxylic acid cycle (TCA), and malate dehydrogenase (MDH), which oxidates malate to generate oxaloacetate (part of TCA). The latter reaction may also occur in the cytosol in the opposite direction, leading to NADH consumption to sustain the malate shuttle towards mitochondria. Inside the mitochondria, NADH is oxidized to NAD+ by complex I (NADH: ubiquinone oxidoreductase) of the electron transport chain, donating its electrons to achieve oxidative phosphorylation and ATP synthesis. Importantly, in eukaryotic cells such as neurons, oxidation of NADH by complex I is the main source of ROS inside the cell [104]. In the cytosol, oxidation of NADH is produced by lactate dehydrogenase (LDH), which regenerates the NAD+ required for glycolysis to proceed. Indeed, the measurement of the NADH/NAD+ ratio may serve as an indicator of the balance between glycolysis and oxidative phosphorylation, which has been used for monitoring real time cellular metabolism [105]. Despite all these metabolic pathways that are present in astrocytes and neurons, both cell types differ in their metabolic profiles. For instance, astrocytes are richer in the expression of lactate dehydrogenase 5 (LDH5), which is better suited to produce lactate from pyruvate. On the contrary, neurons express more LDH1, which is more efficient at consuming lactate to produce pyruvate. These complementary molecular signatures are compatible with lines of evidence showing that neurons “outsource” glycolysis to astrocytes. As such, astrocytes behave as net sources of lactate, while neurons are net sinkers of this metabolite [106,107,108,109]. Importantly, cellular metabolism seems to be highly plastic and under some conditions, neurons can directly use glucose to perform glycolysis and all the subsequent metabolic steps [110,111].

The major cytosolic source of NADPH is the pentose phosphate pathway (PPP), which leads to the oxidative decarboxylation of glucose-6-phosphate (G6P) to produce NADPH and the ribose-5-phosphate sugar required for the synthesis of DNA and RNA [112]. The provision of NADPH obtained by neurons through PPP is relevant under oxidative stress. Indeed, it has been claimed that neurons may increase survival under oxidative stress conditions by diverting the metabolic flux of glucose from glycolysis to PPP in order to produce more NADPH and antioxidant power [113]. In addition, the subcellular levels of NADPH are replenished from the NADH pool by the action of the mitochondrial nicotinamide nucleotide transhydrogenase (NNT) [114]. Indeed, it has been estimated that half of the mitochondrial NADPH in the brain depends on the activity of NNT and interrupting its function may cause oxidative stress [99,115]. The abundance of NADPH is also partially determined by cytosolic as well as mitochondrial kinases (NAD kinases), which convert NAD+ into NADP+. In addition, two enzymes from the TCA cycle reduce NADP+ to NADPH inside the mitochondria, namely mitochondrial isocitrate dehydrogenase 2 (IDH2) and malic enzyme (ME1). Nevertheless, in the cytosol, there is another isocitrate dehydrogenase (IDH1) usually catalyzing the reaction in the opposite direction.

In general, while NADH levels are directly implicated in ATP and ROS synthesis, those of NADPH are directly involved in cellular antioxidant response and also in free radical generation by the enzyme NADPH oxidase [116]. However, given the metabolic conditions of brain cells, the role of NADPH would be predominantly antioxidant [99]. Accordingly, NADPH is used by glutathione reductase to reduce oxidized glutathione, and by thioredoxin reductase to reduce oxidized thioredoxin, which are major components of cellular ROS defense [117]. As both cytosolic and mitochondrial NADPH levels tightly depend on those of NADH, it follows that the concentration of both nucleotides determine ROS defense. Accordingly, it has been shown that the provision of NADH is required to support proper detoxification of peroxide from liver cells by mitochondria [117]. The main roles of NADH and NADPH in cell metabolism and antioxidant pathways are summarized in Figure 4.

Measuring NAD metabolism is of interest due to NAD’s biological importance, and ties to human disease and normal aging. Different methods have been used to determine NAD metabolism. Some of them are highly sensitive, such as liquid chromatography mass spectrometry (LC/MS/MS). However, this approach only gives static information of a population of cells and is also invasive and destructive. Table 2 indicates some advantages and disadvantages of different methods for quantifying NAD metabolism, highlighting the relevant contribution of FLIM.

## 5. Analysis of NADH Autofluorescence by FLIM

It has been known for several decades that NADH emits autofluorescence and, in contrast, NAD+ does not [26]. It is important to notice that, as the spectral properties of NADH fully overlaps with those of NADPH, it is common to measure the fluorescent contribution of both components and denominate them as NAD(P)H. Conversely, reduced flavin adenine dinucleotide (FADH2) does not produce autofluorescence compared to its oxidized version (FAD) [130]. This inverse relationship has been used to measure a “redox ratio” defined as the total fluorescence intensity of FAD divided by the total fluorescence intensity of NADH [131]. As such, under relatively constant FAD, lower levels of NAD(P)H may indicate a larger redox ratio and may correlate with a more oxidative cellular environment.

Complementing the classical intensity-based fluorescence methods, the time-resolved decay of fluorescence by FLIM provides unique information about the environment of fluorophores, including changes in pH, viscosity, or binding state to enzymes [132,133,134,135]. Importantly, at least two configurations and fluorescence lifetimes of NADH can be distinguished with this approach, namely free NADH and protein-bound NADH [32]. This is possible because the fluorescence decay of NADH in solution markedly differs when binding to different proteins, i.e., enzymes. As such, when NADH is in solution (free NADH) it exists in a folded configuration, which causes quenching of the reduced nicotinamide by the adenine group and shortening of its fluorescent lifetime. On the contrary, protein-bound NADH has an extended configuration, favoring a prolonged decay of its fluorescence. As such, the reported lifetime of free NADH in solution is significantly lower (~0.4 ns) than the protein-bound conformation (the lifetime of NADH bound to LDH is 3.4 ns) [136]. Furthermore, taking advantage of their binding to different metabolic enzymes, it has been possible to measure the particular contribution of NADH and NADPH separately by FLIM [99,137]. This may constitute a great diagnostic tool to monitor oxidative stress as NADPH is an element directly involved in redox management.

Different methods can be used to calculate the fluorescence lifetime. For this purpose, data can be fitted into a single-exponential or multi-exponential decay function where the exponential factor tau (τ) corresponds to the fluorescence lifetime of the fluorophore. Nevertheless, it is often not possible to determine the best method to fit the data a priori. As a way to circumvent this limitation, data can be also analyzed by phasor approach. Phasor analysis is a fit-free technique in which the fluorescence decay from each pixel is transformed into a point in a two-dimensional (2-D) phasor space. As such, it works on the unbiased raw data without any approximation, and it does not require a priori knowledge of the sample being imaged, giving instantaneous results. Importantly, FLIM is compatible with confocal or multiphoton laser scanning microscopy as well as wide-field illumination. To obtain more details from each methodology, readers may refer to the following excellent publications [138,139,140,141].

## 6. Potential Monitoring of AD Progression through NADH FLIM

Cumulative evidence from patients, as well as cellular and animal models, have suggested that analyzing the content of NADH and NADPH may be useful to monitor AD progression and oxidative stress. Accordingly, mass spectrometry analysis of brains from triple-transgenic mice (3xTg-AD) showed that this AD model is associated with lower number of metabolites from NAD(P)+/NAD(P)H-dependent reactions [142]. In coherence with this result, it was reported that the brain cortex of 3xTgAD/Polβ+/− mice (in which DNA damage is further exacerbated) has reduced NAD+/NADH ratios [143]. The underlying cause of decreased NAD+/NADH ratio might be explained by an increase in oxidative stress due to PARP the activation. Accordingly, it is expected that the consumption of NAD+ by PARP rises under high oxidative stress and DNA damage [144]. The potential mechanistic relevance of PARP activation during AD pathogenesis has been partially supported by experiments in cultured hippocampal astrocytes treated with β-amyloid, which further activated PARP, while decreasing NAD(P)H autofluorescence as well as mitochondrial oxygen consumption [145]. Furthermore, exogenous treatment of AD patient-derived fibroblasts with NAD, which not only restores NAD+ levels but also inhibits PARP, decreased oxidative stress manifested as a rise in 8-Hydroxy-2′-deoxyguanosine (DNA oxidative damage) and mitochondrial ROS [143]. This is highly similar to what our group has previously reported by showing that the inhibition of PARP-1 reduces H_2_O_2_-induced cell death in MCI and AD lymphocytes [67]. Together, these results suggest that under high oxidative stress conditions manifested during AD, a PARP-mediated decrease in NAD+ content could be sensed by label-free microscopy as a drop in either free/protein-bound NADH or NADPH levels. In support of this possibility, it was determined by FLIM that cultured hippocampal neurons from both 3xTg-AD as well as aged mice have diminished cytoplasmic and mitochondrial concentrations of free NADH, which is the direct source of electrons for the mitochondrial complex I [146]

In a complementary approach that supports the potential relevance of a diagnostic tool based on FLIM, it has been shown that cultured neurons from 3xTg-AD mice manifest an early oxidized redox state and lower GSH defense before macromolecular ROS damage is evident [29]. Strikingly, this oxidative damage was reflected in lower resting levels of NAD(P)H/FAD fluorescence ratio and was fully reversible through treatment with NAM. Interestingly, it has been proposed that NAM, as well as other PARP-1 inhibitors, may be used as a treatment for AD at early stages [103]. In order to further test this therapeutic chance, it would be interesting to analyze the content of NADPH in AD patient-derived ONPs by FLIM during the treatment with PARP-1 inhibitors.

The content of NAD+ and NADH in the aging human brain have been non-invasively evaluated by means of magnetic resonance (MR)-based in vivo NAD assay [129]. In coherence with a progressive loss of mitochondrial activity and lower oxidative stress management during normal aging, an age-dependent decline in the content of NAD, NAD+, and NAD+/NADH ratio coupled to increased levels of NADH was revealed in healthy elderly subjects [129]. Interestingly, the decline in NAD+ levels during human aging has been linked to the development and progression of age-related diseases such as AD [147]. Thus, decreased NAD+ levels associated with aging and neurodegeneration are strikingly compatible with the results observed in AD transgenic mice (described above). The limited information from AD patients in this field, despite the promising results in animal models, stresses the need to improve our knowledge of the disease by using patient-derived cellular models.

We sustain that analyzing AD patient-derived ONPs through NADH FLIM is a valuable approach based on the following arguments. First, oxidative stress is an early feature of AD which is manifested in the olfactory system as well as in cultured patient-derived ONPs. Accordingly, patient-derived ONPs are cells of neuronal lineage and can be easily cultured and non-invasively isolated, constituting a cost-effective way to obtain significant amounts of biological material. Second, the use of NADH and NADPH autofluorescence enables the non-invasive imaging of biological samples, minimizing the perturbation of normal physiological conditions and in a less time-consuming manner. With this approach, AD-related oxidative stress could be sensed as an increased FAD/NAD(P)H ratio or reduced levels of NADH or NADPH, which sustain the synthesis of cytosolic and mitochondrial antioxidant molecules. For all these measurements, FLIM not only provides the exclusive technology to discriminate between NADH and NADPH autofluorescence, but also enables obtaining a higher discrimination between the cytosolic and mitochondrial contribution [99]. Thus, we consider that analyzing AD patient-derived ONPs through NADH FLIM has a great translational potential.

## 7. Perspectives and Future Directions

Label-free monitoring of oxidative stress in patient-derived ONPs may accelerate the discovery of molecules for effectively targeting AD. In this sense, imaging the dynamics of NAD(P)H intrinsic fluorescence (e.g., by FLIM) may offer a readily available, less toxic, and comparatively richer lecture of drug effects compared to classic proteomic and cell-fixation methods. Interestingly, patient-derived ONPs have already been used for drug screening. In particular, these cells were used to test drugs that restored acetylated tubulin patient-derived stem cells with a variety of *SPAST* mutations in Hereditary Spastic Paraplegia (HSP) [48] and to perform a multidimensional phenotypic screening with different natural products in Parkinson’s disease [148].

Different cellular AD models have been used for high-throughput screening (HTS) of therapeutic molecules [149,150,151,152]. For example, a search for inhibitors of calpain activity (to prevent Aβ-induced neurotoxicity) was performed on a library of approximately 120,000 compounds and tested on differentiated SH-SY5Y cells [153]. In another approach, the motility and proliferation of PC12 cells was assessed to test multiple drugs based on Chinese herbal compounds targeting Aβ42-induced apoptosis [154]. Similarly, it has been found that a combination of bromocriptine, cromolyn, and topiramate has a potent anti-Aβ effect on patient-derived iPSCs neurons [155]. Thus, it seems plausible to perform HTS of therapeutic molecules in patient-derived ONPs coupled to label free microscopy.

A potential therapeutic strategy, which could be monitored in patient-derived ONPs, is to delay the AD-associated depletion of free NADH. This is supported by the recent observation that imposed manipulation of cysteine/cystine (Cys/CySS) redox state was able to restore mitochondrial levels of free NADH to normal ranges in neurons from triple transgenic AD-like mice [28]. Given the relevance of free NADH inside cells—not only for redox management but also for metabolic supply, to sustain ATP levels—testing for antioxidant compounds capable of modulating free NADH deserves to be further studied. It is surprising to realize that the use of patient-derived ONPs to study the role of oxidative stress during AD has been, to some extent, neglected during the past decade. At least two reasons may have contributed to this delay; the first is that culturing patient-derived ONPs from biopsies is relatively more difficult and the second is the potential lack of technologies efficient enough to detect subtle changes of oxidative stress. Nevertheless, as highlighted in this article, both reasons can no longer be sustained.

Antioxidant therapies directed against AD have shown limited success; however, they still hold great promise and room for improvement. Some clinical trials in which AD patients were supplemented with antioxidants such as vitamins C and E, either alone or in combination with cholinesterase inhibitors, have failed to improve cognitive function [156,157]. However, other attempts have shown to be moderately effective; as is the case for polyphenols, a group of phytochemicals that showed a great antioxidant and anti-inflammatory potential together with neuroprotective properties [158,159]. As such, clinical trials have suggested that polyphenolic compounds such as curcumin, resveratrol, and green tea catechins may prevent and treat some forms of dementia [160,161,162,163]. Nevertheless, other reports show poor effects of antioxidants on cognitive function, which could be related to their low bioavailability [164,165,166]. Emerging evidence suggests that the combined intervention of different antioxidants may improve therapeutic efficacy. For example, some clinical trials have reported cognitive improvements in AD patients treated with a mix of antioxidant compounds harboring α-tocopherol, NAC, folate, acetyl-L-carnitine, vitamin B12, and S-adenosyl methionine [167]. In line with these findings, fibroblasts derived from AD patients have shown decreased mitochondrial oxidative stress after treatment with lipoic acid and N-acetyl-cysteine (NAC) [168]. It would be extremely interesting to monitor the intrinsic fluorescence of NADPH (reflecting the antioxidant capacity of the cell) in patient-derived ONPs in response to different mixes, proportions, and doses of these antioxidant compounds. Table 3 resumes some candidate natural and chemical compounds that could be successful in clinical trials evaluating them with AD-derived ONPs.

Human embryonic stem cells (ESCs) and subsequently human induced pluripotent stem cells (iPSCs) have emerged as powerful tools due to their ability for modeling neurodegenerative diseases [179]. For instance, three-dimensional (3D) organoids using patient-derived induced pluripotent stem (iPS) cells can recapitulate microcephaly that has been difficult to model in mice [180]. On the other hand, 3D advanced culture models of the brain including blood–brain barrier (BBB) allow a precise study of candidate drugs by recapitulating the brain environment [181]. In this sense, the implementation of a human brain microvessel-on-a-chip that is amenable for quantitative live 3D fluorescence analysis with high-resolution will facilitate the monitoring of NADPH movement and permeability during oxidative stress [182]. Moreover, 3D models can be harnessed to perform cutting-edge super-resolution microscopy, including high resolution volumetric imaging using Focused Ion Beam Scanning Electron Microscopy (FIB-SEM) and also with the novel modality of expansion microscopy, which integrates with lattice light-sheet microscopy (Ex-LLSM) [183]. Thus, these emerging systems to model BBB will significantly improve drug discovery.

The olfactory system, the olfactory ensheathing cells (OECs), has been cultured in three dimensions to understand their behavior in a hampered environment, such as a spinal cord injury [184]. Although there is no evidence of ONPs cultured in 3D, the findings in other cells of the olfactory system suggest ONPs will have the same outcome. Thus, it would be extremely interesting to generate a 3D model for AD with ONPs from patients, incorporating a BBB microfluidic platform and analyzing cell metabolism by label-free microscopy in response to drug treatment like that which was reported in the organotypic microfluidic breast cancer model [185]. This approach will enable us to evaluate both the effect and the efficiency to traverse BBB of the drug candidate in an AD model.

In all, non-invasively isolated ONPs from AD patients coupled to real-time monitoring of relevant metabolic intermediaries may provide an unprecedent platform for early diagnosis and drug discovery. Furthermore, cellular models derived from patients might be sensitive enough to even develop personalized therapies, as has been suggested [186]. Proposed innovations are schematized in Figure 5. We envision that these strategies may generate large improvements required for the timely diagnosis and treatment of this devastating disease.

## Figures and Tables

**Figure 1 ijms-22-06311-f001:**
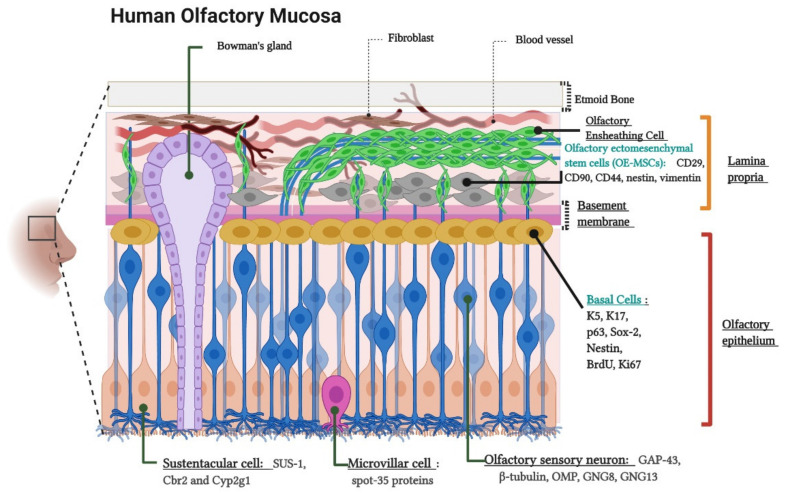
Cytoarchitecture and cellular components of the human olfactory mucosa. Lamina propria components. Olfactory Ensheathing Cells, Bowman’s gland and Olfactory Ectomesenchymal Stem Cells (OE-MSCs). The image indicates the OE-MSCs markers: CD29, CD90, CD44, Nestin, and Vimentin. Olfactory epithelium components. Basal Cells, Olfactory sensory neurons (OSNs) or Olfactory receptor neurons (ORNs), Sustentacular cells, and Microvillar cells. The figure shows basal cell markers: K5 (Keratin 5), K17 (Keratin 17), p63, Sox-2 (SRY-Box Transcription Factor 2), Nestin, BrdU (Bromodeoxyuridine), and Ki-67; ORNs markers: GAP-43 (Growth Associated Protein 43), β-tubulin, OMP (Olfactory Marker Protein), GNG8 (Guanine Nucleotide-binding protein subunit Gamma), and GNG13 (Guanine Nucleotide-binding protein G(I)/G(S)/G(O) subunit Gamma-13)); sustentacular cell markers (SUS-1, Cbr2 (Carbonyl Reductase 2) and Cyp2g1 (Cytochrome P450, family 2, subfamily G, polypeptide 1)) and, microvillar cell marker: (spot-35 proteins). Created with BioRender.com.

**Figure 2 ijms-22-06311-f002:**
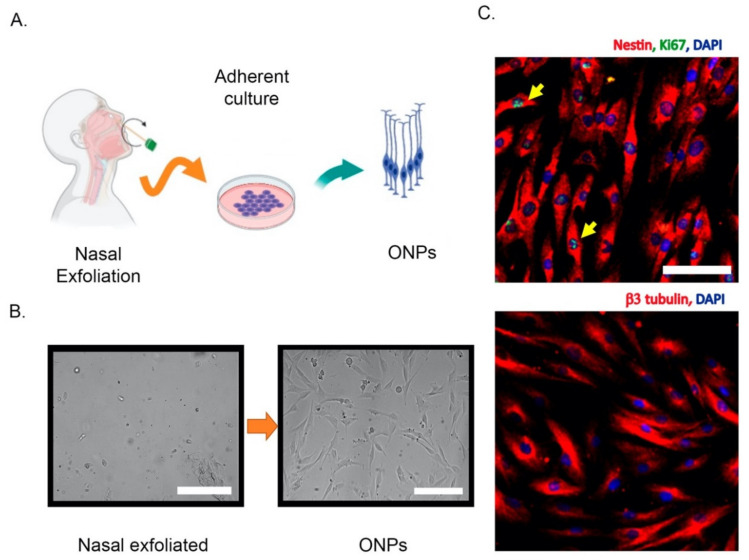
Non-invasive isolation of olfactory neuronal precursors (ONPs). (**A**) Schematic cartoon of the isolation protocol based on the extraction of nasal exfoliate with the subsequent adherent culture and enrichment of ONPs. (**B**) Left, the nasal exfoliate is directly seeded on adherent plates, showing a mixture of cell morphologies. Right, after 1–2 weeks ONPs dividing colonies are easily observed with their characteristic morphologies. (**C**) Upper panel, immunofluorescence of cultured ONPs, depicting the stem cell marker Nestin and Ki67 (yellow arrows) to show active cell proliferation. Lower panel, cultured ONPs express neuronal markers such as β3 tubulin. Cell nuclei are shown by DAPI staining. All scale bars = 100 μm. All images were generated in our lab. Created with BioRender.com.

**Figure 3 ijms-22-06311-f003:**
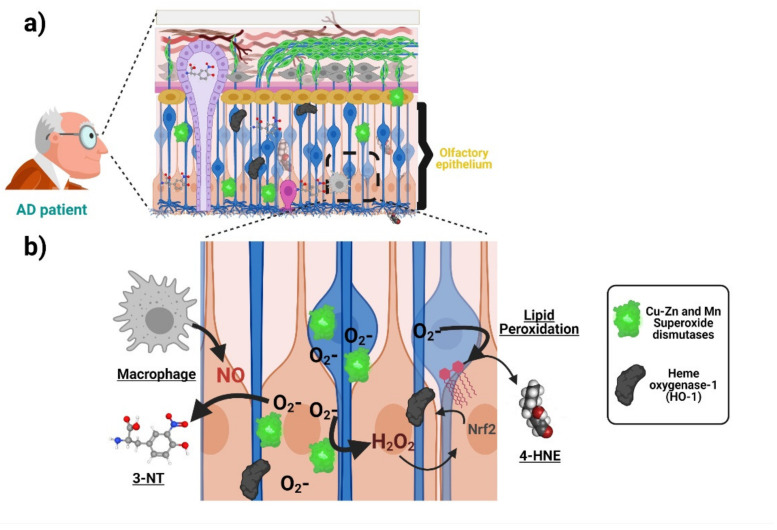
Oxidative stress associated with AD in the olfactory neuroepithelium. (**a**) ONPs and sustentacular cells in the olfactory epithelium (OE) show an increased antioxidant defense with elevated levels of manganese and copper-zinc superoxide dismutases as well as heme oxygenase-1 due to increased oxidative stress in AD patients compared with age-matched controls. Moreover, there is an increase in 3-nitrotyrosine (3-NT) and 4-hydroxynonenal (lipid peroxidation indicator) levels, suggesting AD-associated oxidative damage. (**b**) The increased generation of superoxide anion activates superoxide dismutases (SOD) as an antioxidant response. The generation of other reactive oxygen species (ROS), such as H_2_O_2_, induces the expression of other antioxidant enzymes (heme oxygenase-1). On the other hand, the accumulation of superoxide anion increases the levels of compounds such as 4-hydroxynonenal (4-HNE). Moreover, the increased levels of 3-NT are produced from the interaction of superoxide anion and nitric oxide (NO), whose probable source is located at activated macrophages in the OE of AD patients. Created with BioRender.com.

**Figure 4 ijms-22-06311-f004:**
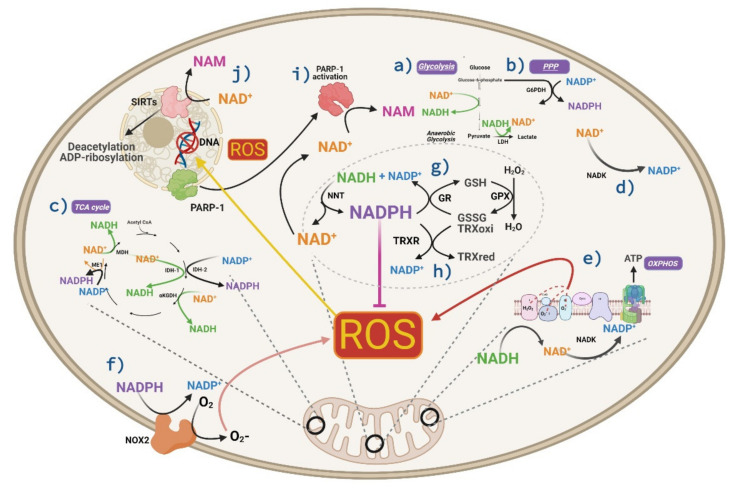
Roles of NADH and NADPH in metabolism and antioxidant pathways. (**a**–**c**) Synthesis of NADH from NAD+ in (**a**) glycolysis, and (**c**) TCA cycle; NADPH from NADP+ in (**b**) PPP and (**c**) TCA cycle. (**d**) Synthesis of NADP+ from NAD+ by NAD+ kinase both in cytosol and mitochondria. (**e**) Oxidation of NADH by complex I is the main source of ROS inside the cell in addition to (**f**) the activation NOX2 that generates ROS through a reduction of oxygen using NADPH as the source of donor electrons. In brain cells, the role of NADPH is predominantly antioxidant; for instance, (**g**) NADPH is used by glutathione reductase to reduce oxidized glutathione, and by (**h**) thioredoxin reductase to reduce oxidized thioredoxin. (**i**) Under oxidative stress and DNA damage, PARP-1 is activated, and this is manifested by an increase in the consumption of NAD+ by PARP. (**j**) On the other hand, the enzymatic activity of SIRTs consumes NAD+. SIRTs catalyze the deacetylation of target proteins by converting NAD+ into NAM. Created with BioRender.com.

**Figure 5 ijms-22-06311-f005:**
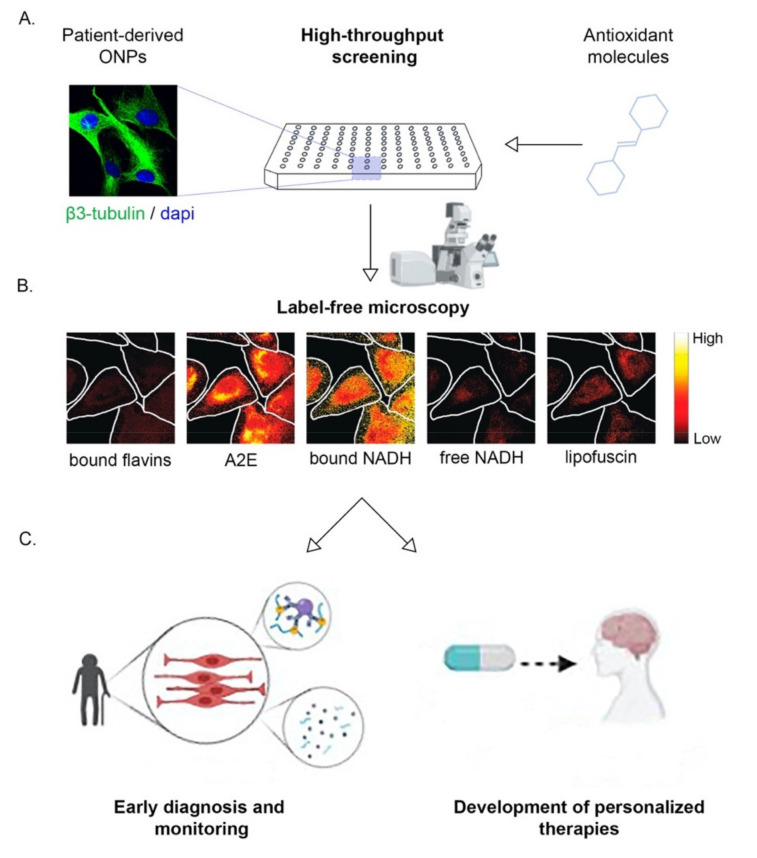
Isolation of patient-derived ONPs coupled to label-free microscopy offers relevant translational outcomes. (**A**) Schematic drawing of the high-throughput screening platform to study different antioxidant molecules. (**B**) Cultured ONPs isolated from control patient-derived neurospheres were analyzed by label-free microscopy, using fluorescence hyperspectral analysis and the intensity of different intrinsic fluorophores was determined [187]. These fluorophores included: bound flavins, fluorescent retinoid derivative bis-retinoid N-retinylidene-N-retinyl ethanolamine (A2E), protein-bound NADH (bound NADH), free NADH, and lipofuscin. The original images from the publication of Gosnell et al. [187] (Figure 2) were adapted (cropped) with permission, following the guidelines of the creative commons license (CC BY 4.0, https://creativecommons.org/licenses/by/4.0/ ). (**C**) Analysis of intrinsic fluorophores such as protein-bound NADH or free NADH could provide relevant translational outcomes such detecting oxidative/metabolic signatures for early AD diagnosis and monitoring. In addition, those the subtle molecular profiling could settle the base for development of personalized therapies to treat AD. Created with BioRender.com.

**Table 1 ijms-22-06311-t001:** Signatures of oxidative stress and other AD mechanistic hypotheses are manifested in patient-derived peripheral cells, iPSCs and ONPs.

Pathogenic Mechanism	Main Finding	Cellular Type	Lineage	References
**Amyloid/Tau**	Platelets from AD patients reproduce the increased amyloidogenic processing of AβPP	Platelets	Non-neuronal	[68]
**Amyloid/Tau**	AD platelets harbor increased levels of a higher molecular weight tau isoform	Platelets	Non-neuronal	[69]
**Amyloid/Tau**	Alteration of AβPP, BACE, and ADAM 10 levels in early stages of the disease	Platelets	Non-neuronal	[70,71,72]
**Amyloid/Tau**	It is suggested a decreased non-amyloidogenic processing of AβPP by a lack of nicastrin mRNA expression in samples obtained from AD patients	Lymphocytes	Non-neuronal	[73]
**Amyloid/Tau**	Altered balance between Aβ-oligomers and PKCε levels in AD. Loss of PKCε-mediated inhibition of Aβ	Fibroblasts	Non-neuronal	[74]
**Amyloid/Tau**	Higher Aβ_42_/Aβ_40_ ratio compared to control cells	*PSEN1* iPSC-derived neural progenitors	Neuronal	[75]
**Amyloid/Tau**	Mutation alters the initial cleavage site of γ-secretase, resulting in an increased generation of Aβ_42_, in addition to an increase in the levels of total and phosphorylated tau	Neuron-derived iPSCs from patients harboring the London FAD *AβPP* mutation V717I	Neuronal	[76]
**Amyloid/Tau**	Oligomeric forms of canonical Aβ impairs synaptic plasticity	Cortical neurons from three genetic forms of AD —*PSEN1* L113_I114insT, *AβPP* duplication (*AβPP*Dp), and Ts21— generated from iPSCs	Neuronal	[77]
**Amyloid/Tau**	Increase in the content and changes in the subcellular distribution of t-tau and p-tau in cells from AD patients compared to controls	Non-invasively isolated ONPs	Neuronal	[9]
**Mitochondria**	Compromise of mitochondrial COX from AD patients	Platelets	Non-neuronal	[78]
**Mitochondria**	Platelets isolated from AD patients show decreased ATP levels	Platelets	Non-neuronal	[79]
**Mitochondria**	AD lymphocytes exhibit impairment of total OXPHOS capacity	Lymphocytes	Non-neuronal	[80]
**Mitochondria**	AD skin fibroblasts show increased production of CO_2_ and reduced oxygen uptake suggesting that mitochondrial electron transport chain might be compromised	Fibroblasts	Non-neuronal	[81]
**Mitochondria**	AD fibroblasts present reduction in mitochondrial length and a dysfunctional mitochondrial bioenergetics profile	Fibroblasts	Non-neuronal	[82]
**Mitochondria**	SAD fibroblasts exhibit aged mitochondria, and their recycling process is impaired	Fibroblasts	Non-neuronal	[83]
**Mitochondria**	Patient-derived cells show increased levels of oxidative phosphorylation chain complexes	Human induced pluripotent stem cell-derived neuronal cells (iN cells) from SAD patients	Neuronal	[84]
**Mitochondria**	Mitophagy failure as a consequence of lysosomal dysfunction	iPSC-derived neurons from FAD1 patients harboring *PSEN1* A246E mutation	Neuronal	[85]
**Mitochondria**	Neurons exhibit defective mitochondrial axonal transport	iPSC-derived neurons from an AD patient carrying *AβPP* -V715M mutation	Neuronal	[86]
**Oxidative Stress**	Increased activity of the antioxidant enzyme catalase in probable AD patients	Erythrocytes	Non-neuronal	[87]
**Oxidative Stress**	Increased production and content of thiobarbituric acid-reactive substances (TBARS), superoxide dismutase (SOD), and nitric oxide synthase (NOS)	Erythrocytes and Platelets	Non-neuronal	[88]
**Oxidative Stress**	Increase in the content of the unfolded version of p53 as well as reduced SOD activity	Peripheral blood mononuclear cells (PBMCs)	Non-neuronal	[89]
**Oxidative Stress**	Exacerbated response to NFKB pathway	PBMCs	Non-neuronal	[90]
**Oxidative Stress**	Increased ROS production in response to H_2_O_2_	PBMCs	Non-neuronal	[66]
**Oxidative Stress**	AD lymphocytes were more prone to cell death after a H_2_O_2_ challenge	Lymphocytes	Non-neuronal	[91]
**Oxidative Stress**	Reduced antioxidant capacity of FAD lymphocytes and fibroblasts together with increased lipid peroxidation on their plasma membrane	Lymphocytes and Fibroblasts	Non-neuronal	[92]
**Oxidative Stress**	Aβ peptides were better internalized and generated greater oxidative damage in FAD fibroblasts	Fibroblasts	Non-neuronal	[93]
**Oxidative Stress**	Aβ peptide caused a higher increase in the oxidation of HSP60	Fibroblasts	Non-neuronal	[94]
**Oxidative Stress**	Reduction in the levels of Vimentin in samples from AD patients	iPSCs-derived neurons from AD patient	Neuronal	[65]
**Oxidative Stress**	Increased levels of hydroxynonenal, Nɛ-(carboxymethyl)lysine), and heme oxygenase-1 in samples from AD patients	Biopsy-derived ONPs	Neuronal	[24]
**Oxidative Stress**	Increased susceptibility to oxidative-stress-induced cell death	Biopsy-derived ONPs	Neuronal	[25]
**ER-Stress**	Impaired ER Ca^2+^ and ER stress in PBMCs from MCIs and mild AD patients	PBMCs	Non-neuronal	[95]
**ER-Stress**	Accumulation of Aβ oligomers induced ER and oxidative stress	iPSC-derived neural cells from a patient carrying *APP*-E693Δ mutation and a sporadic AD patient	Neuronal	[96]
**ER-Stress**	Aβ-S8C dimer triggers an ER stress response more prominent in AD neuronal cultures where several genes from the UPR were upregulated	iPSC-derived neuronal cultures carrying the AD-associated *TREM2* R47H variant	Neuronal	[97]
**ER-Stress**	Accumulation of Aβ oligomers in iPSC-derived neurons from AD patients leads to increased ER stress	iPSC-derived neurons from patients with an *AβPP*-E693Δ mutation	Neuronal	[98]

**Table 2 ijms-22-06311-t002:** Methods for measuring NAD+ and derivatives.

Assay	Analyte	Advantages	Disadvantages	Ref
**Luminometric analysis**	NAD+, NADH, NADP+, and NADPH concentration	Method is reproducible and reported in tissues and cells.	Partial inactivation of luciferase system. Invasive and destructive.	[118]
**Colorimetric Assay using thiazolyl blue**	Intracellular NAD+ concentration	Identifies biological trends that are highly reproducible in the literature.	Indirect measurement affected by minor variations in temperature and pH. Cannot detect low picomolar levels. Invasive and destructive.	[119,120]
**BRET-based biosensors**	NAD+ concentration	Quantifies NAD+ levels in cell culture, tissue, and blood samples. The readout can be performed by a microplate reader or a simple digital camera. Minimum consumption of biological samples.	Invasive and destructive.	[121]
**Reverse phase HPLC**	Endogenous intracellular and extracellular levels of NAD+ and related metabolites	The method uses elements to increase sensitivity.	Limited to low micromolar detection levels. Since many NAD-related metabolites can be converted to one or more metabolites the identified concentrations may be fraught with inaccuracies. Invasive and destructive detection. Static information of a population of cells.	[122]
**LC-MS/MS**	Endogenous intracellular and extracellular levels of NAD+ and related metabolites	High specificity and sensitivity.	The assay requires time, many preparations, and materials not readily available. Static information of a population of cells. Invasive and destructive detection.	[123,124]
**LC-MS/MS (NAD metabolite isotopic labels)**	Endogenous intracellular and extracellular levels of NAD+ and related metabolites	The method provides greater resolution and lower limit of detection.	Static information of a population cells. Invasive and destructive.	[125,126]
**Fluorescent imaging with metabolite sensors**	NADH, NAD+ concentrations, and their ratio	Metabolite sensors may be used to profile metabolic states of living cells in real-time and with single-cell or even subcellular resolution.	Invasive (metabolite sensors are introduced into any cell or organism). With some sensors, fluorescence is sensitive to pH. Other sensors have a limited dynamic range in fluorescence.	[127,128]
**Novel MRI-based process**	NAD+ and NADH concentrations	Non-invasive and non-destructive, measured in healthy aged human brains.	Only measures 2 analytes.	[129]
**Fluorescence Lifetime Imaging (FLIM)**	NAD+, NADH, NADP+, and NADPH	Non-invasive and non-destructive using autofluorescence intensity. May be used to profile metabolic states of living cells in real-time.	Requires an expensive equipment.	[99]

**Table 3 ijms-22-06311-t003:** Natural and chemical compounds that may target ONPs.

Compounds	Targeting	Mechanism	In Vitro/In Vivo Models	Comments	Refs
**Incensole acetate (IA)**	Oxidative stress induced by Aβ	Increased levels of the antioxidant enzyme HO-1	Human olfactory bulb neural stem cells (hOBNSCs)	The cellular model belongs to the olfactory system; therefore, we envision similar results in our proposed cellular model.	[169]
**Curcumin loaded polymeric or lipid nanosuspensions**	Oxidative stress	Elevation of total cellular glutathione levels and enhanced cell viability under oxidative stress	Normal and hypoxic olfactory ensheathing cells (OECs)	The use of OECs (non-myelinating glial cells that wrap olfactory neurons) in hypoxic conditions enables a roadmap to improve the delivery of antioxidants through the nose-to-brain route.	[170,171,172]
**Saturated medium-chain fatty acid (MCFA) decanoic** **acid (10:0)**	Oxidative stress	Upregulation of catalase activity and increase in mitochondrial citrate synthase	Neuroblastoma cells (SH-SY5Y cells)	MCFA decanoic acid has only been evaluated in human cell lines. These findings suggest it is worth testing them in AD patient-derived ONPs.	[173,174]
**Scutellarin (SC)**	Oxidative stress and apoptosis	Enhances the levels of superoxide dismutase	L-Glu-treated HT22 cells/ AD mice induced by AlCl_3_ and D-gal	SC has shown antioxidant and antiapoptotic properties only in induced models of AD; thus, it would be interesting to evaluate these properties in a cellular model derived from AD patients.	[175]
**Curcumin and** **Vitamin D3**	Oxidative stress	Increased SOD enzyme activity and catalase enzyme expression	Primary neuronal cortical culture from rats treated with Aβ	Antioxidant combinations show a synergistic effect that could be tested in an ONP model.	[176]
**TM-10 (a ferulic acid derivative and a highly selective BuChE inhibitor)**	Oxidative stress, Aβ aggregation, butyrylcholinesterase (BuChE) inhibition	Neuroprotective effect against Aβ_42_- mediated SH-SY5Y neurotoxicity, and autophagy induction. In mice, improves scopolamine-inducedmemory impairment	SH-SY5Y cell, U87 cell, AlCl_3_-induced zebrafish AD model, and mice treated with scopolamine	The search of Multi-Target-Directed-Ligands (MTDLs) has allowed fusing novel natural antioxidants derivatives and highly selective BuChE inhibitors. Thus, compounds with multiple biological activities are obtained, including ChE inhibitory activity, MAOs inhibitory potency, antioxidant activity, disaggregation effect on Aβ, and the ability to cross the blood−brain barrier. The use of AD patient-derived ONPs could be a valuable tool for validating these compounds in humans.	[177,178]

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
