# Peer review of "Analyzing Olfactory Neuron Precursors Non-Invasively Isolated through NADH FLIM as a Potential Tool to Study Oxidative Stress in Alzheimer’s Disease"

_ijms, 2021, doi:10.3390/ijms22126311_

Round 1
Reviewer 1 Report
This is a good paper. I have a few suggestions:
- Another figure may be added on ONPs and oxidative stress related pathways.
- Any genetic factors are available, which could be related between AD and olfactory system?
- I would add a little more detailed on natural and chemical compounds, which may target ONP, and could be successful. A chapter or a table on this topics would be nice.
Reviewer 2 Report
Thank you for the opportunity to review this manuscript entitled ‘Analyzing olfactory neuron precursors non-invasively isolated 2 through NADH FLIM as a potential tool to study oxidative 3 stress in Alzheimer’s disease’ by Laura Gómez-Virgilio and co-authors.
The manuscript is well written and logical and presents review of current scientific evidence appropriate for publication in IJMS in present form.
Author Response
R: Thank you so much for this kind appreciation we sincerely hope to make a significant contribution to IJMS.
Reviewer 3 Report
Dear Editor,
The manuscript by Gómez-Virgilio et al. reviews the translational potential of analyzing patient-derived olfactory neuronal precursors lifetime imaging microscopy (FLIM) non-invasively isolated through NADH FLIM to reveal AD-related oxidative stress.
The review is comprehensive and informative. Authors were successful in providing some well compiled opinions and summaries. The mechanistic figures and adding some suggestions for future directions in the conclusion section can be a good starting point for future studies and will be of interest for IJMS readers and beyond.
However, there is a number of major and minor points that would need to be addressed in order to improve the quality of this paper before it can be accepted for publication:
-Line 15: “However; despite the significant knowledge accumulated”. This needs to be rephrased to something like “Lines of evidence”.
-The Introduction lacks a brief mention regarding the essential role of glial cells in oxidative stress before zooming in to focus on neurons. Glial cells, particularly astrocytes, appear to play critical and interactive roles especially at the BBB and BSCB. A recent work by Kitchen et al Cell 2020, has showed that ischemia and hypoxia affect BBB and result in CNS edema and targeting them at the site where they communicate with endothelial cells can be a viable therapeutic target. Authors need to mention this and the role of other glial cells (especially pericyte) in the introduction or at the discussion. References to be included:
- https://pubmed.ncbi.nlm.nih.gov/32413299/
- https://pubmed.ncbi.nlm.nih.gov/16674981/
Also, authors need to mention the role of astrocytes in the formation of tripartite synapse as established by Araque et al. since this has provided the foundation for the role of astrocytes in oxidative stress. Referece:
https://pubmed.ncbi.nlm.nih.gov/10322493/
https://pubmed.ncbi.nlm.nih.gov/19615761/
-Table 1 “Biopsy-derived ONPs” shouldn’t be in bold.
-Line 238-253: This is a rather simplistic overview. Authors need to mention the astrocyte‐neuron lactate shuttle (ANLS) hypothesis postulated in 1994 (Pellerin and Magistretti 1994). According to this, astrocytes serve as a ‘lactate source’ whereas neurons serve as a ‘lactate sink’. Moreover, the opposition by Bak and colleagues who argued that oxidative metabolism of lactate within neurons only occurs during repolarization (and in the period between depolarizations) rather than during neurotransmission activity. The emerging role of astrocytes has helped in settling this debate in favour for ANLS hypothesis. References to be included:
https://pubmed.ncbi.nlm.nih.gov/31318452/
https://pubmed.ncbi.nlm.nih.gov/19393013/
https://pubmed.ncbi.nlm.nih.gov/7938003/
-Line 217-218 “Different cellular AD models have been used for high-throughput screening (HTS) of therapeutic molecules. Authors need to provided references at the end of this sentence. The potential use of HTS in neurodegenerative diseases has recently been reviewed by Aldewachi et al 2021 and also Del Palacio et al 2016. References to be included:
https://pubmed.ncbi.nlm.nih.gov/33672148/
https://pubmed.ncbi.nlm.nih.gov/26962874/
- Towards the end of discussion: authors need to briefly discuss future directions following towards the end of their discussion and conclusion. This could include, but not limit to, the use of humanized self-organized models, organoids, 3D cultures and human microvessel-on-a-chip platforms especially those which are amenable for advanced imaging since they enable real-time monitoring fluorescently labelled NADPH movement and permeability during oxidative stress. References to be included:
https://pubmed.ncbi.nlm.nih.gov/30165870/
https://pubmed.ncbi.nlm.nih.gov/33117784/
https://www.ncbi.nlm.nih.gov/pmc/articles/PMC3817409/
Best
Round 2
Reviewer 1 Report
Authors fulfilled my suggestions
Author Response
We appreciate your suggestions. We sincerely hope to make a significant contribution to IJMS.
Reviewer 3 Report
Dear Editor,
The authors have successfully addressed the majority of my comments in order to improve the quality of the manuscript.
I do believe that the corrections, additional sections and updated references, have contributed to enhancing the clarity of the manuscript, which I can endorse for publication following the successful completion of the below mentioned minor edits.
All the best!
Minor:
-Lines 521-523 "On the other hand, 3D advanced culture models of the brain, mimicking the blood-brain barrier (BBB), allow a precise prediction of the effects of candidate drugs by recapitulating the brain environment [173].". Authors didn't mention the 3D cultures of human microvessel-on-a-chip platforms especially those which are amenable for advanced imaging since they enable real-time monitoring fluorescently labelled NADPH movement and permeability during oxidative stress in live time. The advantage of the open chip design by Salman et al is the ease for chemical fixation and sample handling of the biological material located within the collagen associated with TEM, with high resolution volumetric imaging using Focused Ion Beam Scanning Electron Microscopy (FIB-SEM) and also with the newly developed modality of expansion microscopy combined with LLSM (Ex-LLSM), which they can allow the studying molecular mechanisms with unprecented resolution. Reference:
https://pubmed.ncbi.nlm.nih.gov/33117784/
-Lines 53-543 "Thus, it would be extremely interesting to generate a 3D model for AD with ONPs from patients, incorporating a BBB microfluidic platform and analyzing cell metabolism by label-free microscopy in response to drug treatment like the reported in the organotypic microfluidic breast cancer model [178]". Reference hasn't been included in the reference list.
